# A Selenophene-Incorporated Metal–Organic Framework for Enhanced CO_2_ Uptake and Adsorption Selectivity

**DOI:** 10.3390/molecules25194396

**Published:** 2020-09-24

**Authors:** Pavel A. Demakov, Sergey S. Volynkin, Denis G. Samsonenko, Vladimir P. Fedin, Danil N. Dybtsev

**Affiliations:** 1Nikolaev Institute of Inorganic Chemistry, Siberian Branch of the Russian Academy of Sciences, Novosibirsk 630090, Russia; demakov@niic.nsc.ru (P.A.D.); volynkin@niic.nsc.ru (S.S.V.); denis@niic.nsc.ru (D.G.S.); dan@niic.nsc.ru (D.N.D.); 2Department of Natural Sciences, Novosibirsk State University, 2 Pirogova St., Novosibirsk 630090, Russia

**Keywords:** porous metal–organic frameworks, selenophene, X-ray diffraction studies, adsorption measurements, gas adsorption selectivity

## Abstract

A new metal–organic coordination polymer [Zn_2_(sedc)_2_(dabco)] (**1se**; sedc^2−^ = selepophene-2,5-dicarboxylate; dabco = 1,4-diazabicyclo[2.2.2]octane) was synthesized and characterized by single-crystal X-ray diffraction analysis. This MOF is based on {Zn_2_(OOCR)_4_N_2_} paddle wheels and is isoreticular to the family of [Zn_2_(bdc)_2_(dabco)] derivatives (**1b**; bdc^2−^ = 1,4-benzenedicarboxylate) with **pcu** topology. The gas adsorption measurements revealed that **1se** shows a 15% higher CO_2_ volumetric uptake at 273 K and 28% higher CO_2_ uptake at 298 K (both at 1 bar) compared to the prototypic framework **1b**. Methane and nitrogen adsorption at 273 K was also investigated, and IAST calculations demonstrated a pronounced increase in CO_2_/CH_4_ and CO_2_/N_2_ selectivity for **1se**, compared with **1b**. For example, the selectivity factor for the equimolar CO_2_/CH_4_ gas mixture at 1 bar = 15.1 for **1se** and 11.9 for **1b**. The obtained results show a remarkable effect of the presence of selenium atom on the carbon dioxide affinity in the isoreticular metal–organic frameworks with very similar geometry and porosity.

## 1. Introduction

Carbon dioxide is a corrosive gas and a major atmospheric pollutant, causing a greenhouse effect. Selective capture and sequestration of carbon dioxide is one of the challenging problems in the modern chemical industry. Typically, a chemical sorption of CO_2_ by amines is highly effective; however, the corrosive nature of the chemosorbents and substantial energy penalties during their regeneration demand new viable alternatives [1,2,3,4]. Porous coordination polymers or metal–organic frameworks (MOFs) are recently emerged materials with the greatest potential for adsorption of gases. Remarkable porosity, sufficient stability, as well as vast versatility of the structural and functional design make this class of porous materials among the most perspective adsorbents for selective CO_2_ capture from natural gas or industrial flue gas [5,6,7,8,9,10]. Modular design is a widely used principle to tailor structural and/or functional properties within a certain type of isostructural MOF family, such as IRMOF-1 [11,12,13], MIL-53 [14,15,16], or UiO-66 [17,18,19], which are typically based on the linear 1,4-benzenedicarboxylate (bdc^2−^) linker. Other than a bent shape, the striking feature of ditopic dicarboxylate linkers based on heterocyclic core (Scheme 1) is the polarity of the core and the polarizability of a heteroatom, enabling additional induced dipole interactions with a substrate. Recently, a number of research groups have shown that the incorporation of 2,5-furanedicarboxylate (fdc^2−^) or 2,5-thiophenedicarboxylate (tdc^2−^) into porous MOF structures imbue such materials with quite unique features such as selective ion adsorption, luminescence sensing [20,21,22], dielectric bistability [23], and gas/vapor uptake [24,25].

A porous coordination polymer [Zn_2_(bdc)_2_(dabco)] (dabco = 1,4-diazabicyclo[2.2.2]octane) is a prototypic compound of a bountiful MOF family with highly tunable structural features [26,27,28,29,30]. We have demonstrated that the incorporation of 2,5-thiophenedicarboxylate instead of the 1,4-benzenedicarboxylate into this MOF enhances both the CO_2_ adsorption uptake and the CO_2_/N_2_ adsorption selectivity of [Zn_2_(tdc)_2_(dabco)] by as much as 50%, compared with the original structure [Zn_2_(bdc)_2_(dabco)] at the same conditions [31]. The single-crystal X-ray diffraction studies, as well as the quantum-chemical calculations unveiled the role of the thiophene moieties in the specific CO_2_ binding via an induced dipole interaction between CO_2_ molecules and the heterocycle. Selenium atoms possess even greater radius and higher polarizability than sulfur. Thus, an enhancement of adsorption properties owing to van der Waals intermolecular interactions between a MOF and CO_2_ or other gaseous substrate should be expected if a selenophene moiety is incorporated into such porous material. The 2,5-selenophenedicarboxylate (sedc^2−^) is the heterocyclic linker, structurally and functionally similar to the 2,5-thiophenedicarboxylate. Except for a handful of examples [32,33], sedc^2−^ has mostly been ignored in MOF chemistry so far. Herein, we report the synthesis and investigation of a new selenophenedicarboxylate-based porous MOF [Zn_2_(sedc)_2_(dabco)] (**1se**), which is isoreticular to both [Zn_2_(bdc)_2_dabco)] (**1b**) and [Zn_2_(tdc)_2_(dabco)] (**1t**). The impact of selenium heteroatom on the adsorption properties of [Zn_2_(sedc)_2_(dabco)] is discussed in detail.

## 2. Results and Discussion

### 2.1. Structure Description

The coordination polymer **1se** was synthesized by the solvothermal reaction of Zn(NO_3_)_2_·6H_2_O, H_2_sedc, and dabco in DMF at 100 °C for 24 h. According to single-crystal X-ray diffraction data, **1se** crystallizes in the tetragonal space group *P*-42_1_c. The asymmetric unit contains two crystallographically independent Zn(II) atoms with the same coordination environment of four O atoms of four bridging carboxylate groups and one N atom of dabco bridge. Zn–O distances lie in the range 2.015(2)–2.062(2) Å. Zn–N distances are 2.040(2) and 2.047(2) Å. Zn(1) and Zn(2) atoms form binuclear ‘paddlewheel’ blocks {Zn_2_(OOCR)_4_} interconnected by four sedc^2−^ anions into square-grid network. The angular shape of the sedc^2−^ ligands results in a considerable distortion of the binuclear blocks (Figure 1a) and corrugation of the square-grid layers (Figure 1b). These layers are bound by dabco bridges to form a three-dimensional porous framework with a primitive cubic topology (**pcu**)—the same as the other MOFs of the isoreticular family [Zn_2_(bdc)_2_(dabco)].

The coordination network in **1se** contains a three-dimensional system of intersecting channels, with the widest channels running along the *c* axis, across the windows of the square-grid layers (see Figure 1b). The apertures of these channels are 5 × 8 Å. Two different types of smaller channels across the main channels of the tetragonal structure have smaller the apertures of ca. 2.5 × 5 Å or 3.5 × 4 Å, depending on the particular arrangement of arks of the sedc^2−^ ligands (Figure 1c). Importantly, all selenium atoms of the heterocycles are immediately accessible for interactions with potential substrate molecules, allowing a clear verification of the hypothesis of this work. We should also point out that the crystal structure **1se** is similar to the known compound [Zn_2_(tdc)_2_(dabco)] (**1t**), based on 2,5-thiophenedicarboxylate [31] although the distortions of the binuclear blocks {Zn_2_(OOCR)_4_} and the square-grid layers in **1t** are even more pronounced as a result of stronger bending of tdc^2−^ ligand [34], compared with that of sedc^2−^.

### 2.2. Characterization and Activation

To evaluate the thermal stability of **1se**, thermogravimetric analysis (TGA, Appendix A) was performed. The first observed 30% weight loss step occurs between 80 °С and 200 °C, referring to the evaporation of four guest DMF molecules per formula unit (calculated: 30%). The metal–organic framework **1se** itself is stable up to ca. 280 °C, after which the irreversible framework decomposition apparently takes place. Such thermal stability is comparable to other reported MOFs containing sedc^2−^ anions (*T*_decomp_ = 250 ÷ 320 °C) [32,33]. Such substantial difference between the temperature of the evaporation of guests and the temperature of the framework decomposition makes it possible direct activation of **1se** by heating in a vacuum, obtaining the guest-free activated framework [Zn_2_(sedc)_2_(dabco)] (**1se****′**). The infrared spectra of both **1se** and **1se****′** (Appendix A) contain the characteristic bands of Csp^2^-H valence vibrations (3073 cm^−1^), Csp^3^-H valence vibrations (2965 cm^−1^ and 2935 cm^−1^), antisymmetric (1590 cm^−1^) and symmetric (1360 cm^−1^) carboxylate group vibrations. Infrared spectrum of **1se** also contains a characteristic band of CO_amide_ stretchings (1666 cm^−1^), which is absent on the spectrum of **1se****′**, confirming a complete desolvation of metal–organic framework **1se** during the activation process. The PXRD data (Figure 2) suggest the phase purity of the synthesized compound **1se**. The powder diffraction pattern of **1se****′** is generally very similar to that of **1se** although there is a noticeable shift of some of the reflexes to lower angles. For example, the (2 0 0) reflection at 2θ = 8.57° (**1se**) is shifted to 2θ = 8.36° (**1se****′**) indicating a slight extension of the metal–organic framework upon its activation. Similar guest-assisted breathing of the framework was observed earlier for **1b** and **1t** [31,35]. The unit cell parameters for **1se****′** were refined according to the powder data using the Powdercell program [36] and provided in Table A2. While the crystallographic parameter *c* is almost intact, the parameters *a* and *b* in **1se****′** are longer by ca. 2.6% than in **1se**, likely due to certain straightening of the sedc^2−^ dicarboxylate bridges. Overall, the unit cell volume is expanded by 5.2% during the activation of the compound.

### 2.3. Adsorption Measurements

The textural characteristics of the evacuated compound **1se****′** were studied by a nitrogen porosimetry at 77 K. The adsorption isotherm (Figure 3) corresponds to the type I with no hysteresis, typical to microporous adsorbents. The pore volume measured at *p*/*p*^0^ = 0.95 is *V*_pore_ = 0.57 cm^3^·g^−1^; the calculated BET surface area is A_BET_ = 1504 m^2^·g^−1^, respectively (see other details in Appendix A). The experimental *V*_pore_ is very much consistent with the theoretically expected value based on a solvent accessible volume of **1se****′**, calculated using PLATON routine [37] (0.58 cm^3^·cm^−3^ or 0.55 cm^3^·g^−1^). The pore-size distribution, calculated by DFT method from the N_2_ adsorption isotherm, gives a value of a pore size near 7 Å, which corresponds to the van der Waals diameter of the large cuboidal cages inside the **pcu** net. For a reference, the pore volume of the activated MOFs based on thiophenedicarboxylic acid (**1t****′**) or benzenedicarboxylic acid (**1b****′**), reported earlier, are 0.68 cm^3^·g^−1^ and 0.75 cm^3^·g^−1^, respectively. The gravimetric porosity directly depends on molecular weight of a compound, which is notably higher for **1se**, than for **1t** and **1b**. The comparison of experimental volumetric porosities for these MOFs results in a more or less comparable values: 0.60 cm^3^·cm^−3^ (**1se****′**), 0.64 cm^3^·cm^−3^ (**1t****′**), 0.62 cm^3^·cm^−3^ (**1b****′**), see also Appendix A. Most importantly, the compounds **1b**, **1t**, and **1se** represent a suitable family of porous materials where the influence of different heteroatoms on the gas adsorption properties could be systematically analyzed and assessed since the other parameters are almost identical. Being motivated by such an opportunity, we investigated the adsorption properties of **1****se****′** towards industrially important gases (CO_2_ and CH_4_) and compared the obtained data with the other prototypes **1b****′** and **1t****′**.

The gravimetric CO_2_ adsorption uptakes at 1 bar for **1se****′** are 110 cm^3^·g^−1^ (273 K, see Appendix A) and 46 cm^3^·g^−1^ (298 K). Such numbers are comparable with the literature data of CO_2_ adsorption by **1b****′** (122 cm^3^·g^−1^ at 273 K, 46 cm^3^·g^−1^ at 298 K). Taking into account the crystallographic densities, the corresponding volumetric uptakes for **1se****′** were calculated to be 116 cm^3^·cm^−3^ (273 K) and 48.5 cm^3^·cm^−3^ (298 K), which exceeds those for **1b****′** by 15% (101 cm^3^·cm^−3^) at 273 K (see Figure 4) and by 28% (38 cm^3^·cm^−3^) at 298 K, respectively, convincingly confirming a positive effect of the polarizable heteroatom (Se) on the absorption properties of porous materials [38,39,40,41]. Similarly, the isosteric heat of CO_2_ adsorption at zero coverage *Q*_st_(0) for **1se****′** (19.9 kJ·mol^−1^) is greater than for **1b****′** (19.0 kJ·mol^−1^), indicating stronger binding of CO_2_ with the porous framework containing the selenophene heterocycle. We must mention here that the CO_2_ adsorption by **1t****′** is still the highest among the MOFs discussed here, both in terms of the gravimetric uptakes (153 cm^3^·g^−1^ at 273 K, 67.5 cm^3^·g^−1^ at 298 K, 1 bar) and volumetric uptakes (143 cm^3^·cm^−3^ at 273 K, 63.1 cm^3^·cm^−3^ at 298 K, 1 bar). The isosteric heat of CO_2_ adsorption by **1t****′** (23.7 kJ·mol^−1^) also suggests that the thiophene moieties have greater impact on the CO_2_ adsorption. It is probably the polarity of the aromatic ring that contributes to a stronger binding between the polar CO_2_ guest and porous MOF host. Based on the experimental data, the following dependence of the CO_2_ uptake on the nature of the dicarboxylate anion was established: tdc^2−^ > sedc^2−^ > bdc^2−^. This dependence strengthens the earlier claim that the substitution of the common terephthalate linkers to heterocyclic ones should enhance the adsorption properties of the MOF material due to induced dipole interactions. In terms of the gas storage, the thiophene-containing tdc^2−^ seems to be an optimal choice for such substitution since the incorporation of heavier sedc^2−^ no longer improves the gas adsorption capacity of the framework.

The CH_4_ and N_2_ adsorption–desorption isotherms for **1se****′** and **1b****′** were measured up to *p* = 1 bar at 273 K. The gas adsorption isotherms are shown on the Figure 5. For **1se****′** the gravimetric adsorption volumes at 1 bar are 16.2 cm^3^·g^−1^ (CH_4_) and 6.0 cm^3^·g^−1^ (N_2_) adsorption, respectively. For **1b****′**, the corresponding uptakes are 20.0 cm^3^·g^−1^ and 6.6 cm^3^·g^−1^. The gravimetric gas uptakes for **1se****′** are slightly lower than for **1b****′**, mainly due to the higher density of the former. On the contrary, the volumetric assessment indicates higher gas adsorption by **1se****′**, than by **1b****′**, supporting the concept of stronger van der Waals interactions of methane/nitrogen with sedc^2−^ than with bdc^2−^.

The sequestration of CO_2_ from N_2_ or from CH_4_ is a critical technology for a reduction of environmental risks and for protection of natural gas pipelines, respectively. The CO_2_/CH_4_ as well as CO_2_/N_2_ gas adsorption selectivity factors were calculated by three commonly employed approaches: (i) as the ratio of the adsorbed volumes at 1 bar (*S_V_*), (ii) as the ratio of Henry constants (*S_K_*), and (iii) by the ideal adsorbate solution theory (*S*_IAST_). The calculated selectivity values are summarized in Table 1, and the details of the calculations are provided in Appendix A. By any criteria used, the incorporation of sedc^2−^ evidently increases the adsorption selectivities of **1se****′**, compared with the prototypic **1b****′**. The calculated CO_2_/CH_4_ selectivity factors for **1se****′** (*S_V_* = 6.8, *S_K_* = 4.8, *S*_IAST_ = 5.6) are comparable or even exceed those reported for other MOFs with promising application potential for separation of such small molecules [42,43,44]. The CO_2_/N_2_ adsorption selectivity factors for **1se****′** are also quite remarkable *S_V_* = 18.6, *S_K_* = 12.9, *S*_IAST_ = 15.1 (Figure 6) for a porous MOF with no unsaturated metal centers. The results obtained for **1se****′** are superior to the CO_2_/N_2_ selectivity factors for both **1b****′** and **1t****′**. Particularly, the IAST CO_2_ adoption selectivity for the equimolar CO_2_ + N_2_ gas mixture for **1se****′** is ca. 25% greater than for **1b****′** or for **1t****′** (see also Appendix A). Such remarkable increase should apparently be attributed to the nature of the heterocyclic moiety since the pore geometry and other structural parameters of the investigated MOFs are, essentially, the same. However, the contribution of sieving effect is also possible, as the channels in **1se** situated along two of three directions have the apertures smaller than 5Å [45,46]. In terms of the CO_2_/N_2_ selective separation, the incorporation of the selenophene-containing anion provides the best performance in the row: sedc^2−^ > tdc^2−^ > bdc^2−^. Moreover, a rather low CO_2_ adsorption enthalpy for **1se****′** (19.9 kJ·mol^−1^) ensures the facile regeneration of the porous adsorbent in a cyclic CO_2_ sequestration process. The unique combination of remarkable CO_2_/N_2_ adsorption selectivity, high CO_2_ uptake, and one of the lowest CO_2_ adsorption enthalpies puts the title MOF **1se****′** among the best porous materials for practical purification of the industrial exhausts.

## 3. Materials and Methods

## 3.1. Reagents

Commercial starting reagents 2,5-selenophenedicarboxylic acid (Angene), 1,4-diazabicyclo[2.2.2]octane (>98.0%, TCI), terephtalic acid (>98.0%, Hidmon), *N*,*N*-dimethylformamide (reagent grade, Vekton) were used as purchased without purification.

## 3.2. Instruments

IR spectra in KBr pellets were recorded in the range 4000−400 cm^−1^ on a VERTEX 80 spectrometer. Elemental (C, H, N) analysis was made on a varioMICROcube analyzer. Powder X-ray diffraction (PXRD) analysis was performed at room temperature on a Shimadzu XRD-7000 diffractometer (Cu-Kα radiation, λ = 1.54178 Å). Thermogravimetric analysis was carried out on a Netzsch TG 209 F1 Iris instrument. The experiments were carried out under He flow (30 cm^3^·min^−1^) at a 10 K·min^−1^ heating rate. Adsorption experiments were performed using Quantachrome Autosorb iQ device. Low-pressure gas adsorption isotherms at 273 K and 298 K were recorded with a thermostat TERMEX Cryo-VT-12 to adjust temperature with 0.1 K accuracy. The database of the National Institute of Standards and Technology was used as a source of *p*−*V*−*T* relations at experimental pressures and temperatures. Elemental (Zn, Se) ICP-MS analysis was carried out using Agilent 8800. The samples of **1se** and **1se****′** were digested in the mixture of HCl 36% water solution and H_2_O_2_ 30% water solution, then diluted by water prior to ICP-MS.

## 3.3. X-ray Crystallography

Diffraction data for single-crystal **1****se** were obtained at 130 K on an automated Agilent Xcalibur diffractometer equipped with an area AtlasS2 detector (graphite monochromator, λ(MoKα) = 0.71073 Å, ω-scans with a step of 0.5°). Integration, absorption correction, and determination of unit cell parameters were performed using the CrysAlisPro program package [47]. The structures were solved by dual space algorithm (SHELXT [48]) and refined by the full-matrix least squares technique (SHELXL [49]) in the anisotropic approximation (except hydrogen atoms). Positions of hydrogen atoms were calculated geometrically and refined in the riding model. The crystallographic data and details of the structure refinement are summarized in Table A1. CCDC 2026693 contains the supplementary crystallographic data for this paper. These data can be obtained free of charge from The Cambridge Crystallographic Data Center at https://www.ccdc.cam.ac.uk/structures/.

## 3.4. Synthetic Procedures

Synthesis of [Zn_2_(sedc)_2_(dabco)]·4DMF (**1se**) 100 mg (0.34 mmol) of Zn(NO_3_)_2_·6H_2_O, 66 mg (0.30 mmol) of H_2_sedc, 20 mg (0.18 mmol) of dabco and 5.00 mL of DMF were mixed in a glass flask. The mixture was hit in the ultrasound bath for 10 min and heated at 100 °С for 24 h. After the cooling to the room temperature white-yellow precipitate was filtered off, washed thrice with DMF and dried in air. Yield: 89 mg (61%). IR spectrum (KBr, cm^−1^) characteristic bands: 3424 (w, br, νOH); 3073 (w, νCsp^2^-H); 2965 and 2925 (w, νCsp^3^-H); 1666 (s, νCO_amide_); 1627 (m, νCOO_as_); 1380 (s, νCOO_s_). PXRD data (Figure 2) confirmed the phase purity of the product. Elemental analysis data calculated for [Zn_2_(C_6_H_2_SeO_4_)_2_(C_6_H_12_N_2_)]4C_3_H_7_NO (%): C, 37.2; H, 4.6; N, 8.7. Found (%): C, 37.0; H, 4.6; N, 8.6. ICP-MS data. Zn:Se molar ratio: 1.02: 1 (theor = 1:1). TGA: 30% weight loss step at ca.150 °C (30% calculated for 4DMF).

Synthesis of [Zn_2_(sedc)_2_(dabco)] (**1se****′**) The sample of **1** was activated by keeping in a primary vacuum (10^−9^ bar) at 50 °C for 2 h, then at 70°C for 2 h, and at last 90 °C for 12 h with 1°·min^−1^ heating and cooling rates. IR spectrum (KBr, cm^−1^) characteristic bands: 3419 (w, br, νOH); 3075 (w, νCsp^2^-H); 2963 and 2926 (w, νCsp^3^-H); 1585 (m, νCOO_as_); 1356 (s, νCOO_s_). PXRD data (Figure 2) confirmed the phase purity of the product. Elemental analysis data calculated for [Zn_2_(C_6_H_2_SeO_4_)_2_(C_6_H_12_N_2_)] (%): C, 31.9; H, 2.4; N, 4.1. Found (%): C, 31.8; H, 2.5; N, 4.5. ICP-MS data. Zn:Se molar ratio: 1.07: 1 (theor = 1:1).

Synthesis of [Zn_2_(bdc)_2_(dabco)]·4DMF·½H_2_O (**1b**) This process carried out according to the published procedure [50]: 300 mg (1.0 mmol) of Zn(NO_3_)_2_·6H_2_O, 165 mg (1.0 mmol) of H_2_bdc, 60 mg (0.54 mmol) of dabco and 18.0 mL of DMF were mixed in a glass flask. The mixture was hit in the ultrasound bath for 10 min and heated at 120 °С for 48 h. After the cooling to the room temperature white precipitate was filtered off, washed thrice with DMF and dried in air. Yield: 380 mg (88%). PXRD data (Appendix A) confirmed the phase purity of the product.

Synthesis of [Zn_2_(bdc)_2_(dabco)] (**1b****′**) The sample of **1b** was activated by keeping in a primary vacuum (10^−9^ bar) at 50 °C for 2 h, then at 80 °C for 2 h, and at 100 °C for 12 h with 1°·min^−1^ heating and cooling rates. PXRD data (Appendix A) confirmed the phase purity of the product.

## 4. Conclusions

To summarize, a new porous metal–organic framework [Zn_2_(sedc)_2_(dabco)], based on 2,5-selenophenedicarboxylate anions (sedc^2−^) was synthesized and characterized. [Zn_2_(sedc)_2_(dabco)] is based on the {Zn_2_(OOCR)_4_N_2_} paddle-wheel blocks and adopts distorted primitive cubic topology, similar to other important representatives of this isoreticular family [Zn_2_(bdc)_2_(dabco)] and [Zn_2_(tdc)_2_(dabco)] (bdc^2−^ = 1,4-benzenedicarboxylate, tdc^2−^ = 2,5-thiophenedicarboxylate). The dependence of the gas adsorption properties on the nature of the bridging anion was systematically investigated. According to the experimental and literature data, the CO_2_ adsorption uptake increases along the following sequence: [Zn_2_(bdc)_2_(dabco)] < [Zn_2_(sedc)_2_(dabco)] < [Zn_2_(tdc)_2_(dabco)]. On the other hand, the CO_2_/N_2_ adsorption selectivity follows the trend [Zn_2_(bdc)_2_(dabco)] < [Zn_2_(tdc)_2_(dabco)] < [Zn_2_(sedc)_2_(dabco)]. Such dependencies strengthen the earlier observation that the incorporation of heterocyclic moieties provides additional adsorption sites which enhances the adsorption properties of the corresponding porous frameworks. The selenophene-based anion seems to have a particular distinction towards CO_2_ molecules. The new porous compound [Zn_2_(sedc)_2_(dabco)] features a unique combination of remarkable CO_2_/N_2_ adsorption selectivity, high CO_2_ uptake, and one of the lowest CO_2_ adsorption enthalpies, which makes it a very promising material for CO_2_ sequestration applications.

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
