# Peer review of "A Selenophene-Incorporated Metal–Organic Framework for Enhanced CO2 Uptake and Adsorption Selectivity"

_molecules, 2020, doi:10.3390/molecules25194396_

Round 1

Reviewer 1 Report

In this paper,author report the synthesis and investigation of a new selenophenedicarboxylate-based porous MOF [Zn2(sedc)2(dabco)]. This frame structure is not very novel. What's interesting is the design idea of the author's experiment. The following places need to be modified.

  1. Please add the 1se checkcif report in the attached materials.
  2. It is recommended to supplement the processing method of the solvent molecule in the crystal structure data and the method for determining solvent molecules in molecular formula.
  3. In the introduction, it is recommended to add a description of compounds containing fdc with similar structures to 1se.
  4. Please pay attention to modify the format of the molecular formula appearing in the references.

Author Response

Reviewer 1:

In this paper,author report the synthesis and investigation of a new selenophenedicarboxylate-based porous MOF [Zn2(sedc)2(dabco)]. This frame structure is not very novel. What's interesting is the design idea of the author's experiment. The following places need to be modified.

1. Please add the 1se checkcif report in the attached materials.

The checkcif report is now attached to the submission supplementary files.

2. It is recommended to supplement the processing method of the solvent molecule in the crystal structure data and the method for determining solvent molecules in molecular formula.

All guest DMF molecules were located directly. The overall structure with the solvent does not contain any void volume after the refinement. Thus, the molecular formula [Zn2(sedc)2(dabco)]·4DMF was first established by single-crystal XRD, then confirmed by CHN, ICP-MS and TG data.

3. In the introduction, it is recommended to add a description of compounds containing fdc with similar structures to 1se.

The fdc-containing isoreticular MOF with the proposed formula [Zn2(fdc)2(dabco)] is not yet synthesized and reported. We suppose that the bending angle of fdc2- is too extreme to form stable network with such pcu topology. However, there are several known examples, when fdc (angle ~ 134°) and bdc (angle = 180°) form isoreticular frameworks, that we already mentioned in the introduction [22-24].

4. Please pay attention to modify the format of the molecular formula appearing in the references. 

We thank the Reviewer for this correction. The reference list is carefully revised.

Reviewer 2 Report

Article is very well written and presented. I have couple of questions.

  1. Selenium is a soft donor. Are there any interference in the metal-framework synthesis. 
  2. What is responsible for the better performance of ZnSedc when it is raw?
  3. Please include the CCDC number in crystallographic data table. I am confused by Table 2 (an incomplete crystallographic info). I see those data belongs to 1Se'. Please provide more if it is a separate single crystal structure data.
  4. There are couple of extra peaks in the PXRD of 1Se' (exp) vs 1Se' (Theor). Any explanation?

Author Response

Reviewer 2:

Article is very well written and presented. I have couple of questions.

1. Selenium is a soft donor. Are there any interference in the metal-framework synthesis. 

Indeed the selenium-containing ligands are known to be very soft bases, however, the selenium atom in selenophene ring, passivated by two electron-withdrawing carboxylate groups, is a very poor donor, which is not able to form stable coordination complexes with Zn(II) in DMF solvent environment. Therefore, we did not observe any interference of Se atom during the syntheses.

2. What is responsible for the better performance of ZnSedc when it is raw?

Firstly, there is a typo in the corresponding sentence (raw à row). The corrected sentence is: In terms of the CO2/N2 selective separation, the incorporation of the selenophene-containing anion provides the best performance in the row: sedc2– > tdc2– > bdc2–.

Secondly, the better performance of 1se¢ in CO2/N2 selective separation should mainly be attributed to the higher polarizability of the selenophene core, compared with thiophene and benzene analogues. The contribution of sieving effect is also possible, as the Se atom has somewhat greater radius, compared with S and, especially, compared with C, which may effectively reduce the aperture of channels. The corresponding argument is included in the revised discussion part. 

3. Please include the CCDC number in crystallographic data table. I am confused by Table 2 (an incomplete crystallographic info). I see those data belongs to 1Se'. Please provide more if it is a separate single crystal structure data.

The CCDC 2026693 number is added to the Table A1 (1se).

We were unable to perform the single-crystal XRD for 1se¢ due to the fragmentation of 1se single crystals upon activation. As we mentioned in the manuscript, the unit cell parameters of 1se¢ were determined from the experimental powder X-ray diffraction data using Powdercell software and the unit cell parameters of 1se as a starting point of the refinement.

4. There are couple of extra peaks in the PXRD of 1Se' (exp) vs 1Se' (Theor). Any explanation?

Several unidentified low intensity reflexes at 2θ ~ 7.75° and 8.85° could indicate a slow reaction of 1se¢ with airborne moisture. However, this process could not affect the adsorption data, as the activation was carried directly in the measuring device and the activated samples did not have to be taken on air before the sorption measurements.

Reviewer 3 Report

The article titled "A Selenophene-Incorporated Metal-Organic Framework for Enhanced CO2 Uptake and Adsorption Selectivity" submitted by Demakov et al. reports a selenium MOF [Zn2(sedc)2(dabco)] for selective CO2 adsorption over CH4 and N2. The adsorption comparison between [Zn2(sedc)2(dabco)] and [Zn2(bdc)2(dabco)] shows a slightly increase for CO2 in [Zn2(sedc)2(dabco)], which was attributed to the binding effect of the selenium atom in current manuscript. But I feel that the sieving effect in the relatively compact [Zn2(sedc)2(dabco)] would make a major contribution for such phenomenon (Chem 2020, 6, 337–363). However, I would glad to see the publication of this work on Molecules, after minor revisions made for follow concerns:

  1. Figure 3, to avoid misleading, the inset pore size distribution is suggested to present in regular pore diameters rather than half pore width.
  2. It is suggested to make a full comparison of pore size, porosity and pore volumes of the mentioned three isoreticular MOFs in this manuscript.
  3. To provide a full perspective of this subject, it is suggested to add some recent progress of relevant separation (e.g. doi: 10.1002/sstr.202000022) in the introduction section.

Author Response

Reviewer 3:

The article titled "A Selenophene-Incorporated Metal-Organic Framework for Enhanced CO2 Uptake and Adsorption Selectivity" submitted by Demakov et al. reports a selenium MOF [Zn2(sedc)2(dabco)] for selective CO2 adsorption over CH4 and N2. The adsorption comparison between [Zn2(sedc)2(dabco)] and [Zn2(bdc)2(dabco)] shows a slightly increase for CO2 in [Zn2(sedc)2(dabco)], which was attributed to the binding effect of the selenium atom in current manuscript. But I feel that the sieving effect in the relatively compact [Zn2(sedc)2(dabco)] would make a major contribution for such phenomenon (Chem 2020, 6, 337–363). However, I would glad to see the publication of this work on Molecules, after minor revisions made for follow concerns:

1. Figure 3, to avoid misleading, the inset pore size distribution is suggested to present in regular pore diameters rather than half pore width.

The authors thank the Reviewer for this suggestion. The scale is now corrected.

2. It is suggested to make a full comparison of pore size, porosity and pore volumes of the mentioned three isoreticular MOFs in this manuscript.

The authors accept the Reviewer’s suggestion. A new Table S2 is added in the revised ESI file and referenced in the main text.  

Table S2. The comparison of porosities between the isoreticular 1t¢, 1se¢ and 1b¢.

Compound

pore diameter, Å

(from the X-ray data)

Vpore, cm3·g–1

(from the X-ray data)

Vpore, cm3·cm–3

(experimental)

1t' [from Ref. 30]

6 x 8

0.68

0.64

1se'

5 x 8

0.55

0.60

1b'

5 x 8

0.75 

0.62

3. To provide a full perspective of this subject, it is suggested to add some recent progress of relevant separation (e.g. doi: 10.1002/sstr.202000022) in the introduction section.

Several new references  (refs. 5, 45, 46 in the revised MS) are added following this comment of the reviewer. 

Although the diameter of channels and the measured pore volumes of porous compounds 1b, 1t and 1se are very similar, the possible contribution of seiving effect in the gas separation performance should not be ignored since the larger diameter of selenium atom could affect the actual aperture of the channels in 1se'. Following the comments of the reviewers we include such possibility in the discussion part of the revised manuscript:

“However, the contribution of sieving effect is also possible, as the channels in 1se situated along two of three directions have the apertures smaller than 5Å [45,46].”
